# AdaFlood: Adaptive Flood Regularization

## Abstract

Although neural networks are conventionally optimized towards zero training loss, it has been recently learned that targeting a non-zero training loss threshold, referred to as a flood level, often enables better test time generalization. Current approaches, however, apply the same constant flood level to all training samples, which inherently assumes all the samples have the same difficulty. We present AdaFlood, a novel flood regularization method that adapts the flood level of each training sample according to the difficulty of the sample. Intuitively, since training samples are not equal in difficulty, the target training loss should be conditioned on the instance. Experiments on datasets covering four diverse input modalities – text, images, asynchronous event sequences, and tabular – demonstrate the versatility of AdaFlood across data domains and noise levels.

## 1 Introduction

Preventing overfitting is an important problem of great practical interest in training deep neural networks, which often have the capacity to memorize entire training sets, even ones with incorrect labels (Neyshabur et al., 2015; Zhang et al., 2021). Common strategies to reduce overfitting and improve generalization performance include weight regularization (Krogh & Hertz, 1991; Tibshirani, 1996; Liu & Ye, 2010), dropout (Wager et al., 2013; Srivastava et al., 2014; Liang et al., 2021), label smoothing (Yuan et al., 2020), and data augmentation (Balestriero et al., 2022).

Although neural networks are conventionally optimized towards zero training loss, it has recently been shown that targeting a non-zero training loss threshold, referred to as a flood level, provides a surprisingly simple yet effective strategy to reduce overfitting (Ishida et al., 2020; Xie et al., 2022). The original Flood regularizer (Ishida et al., 2020) drives the *mean* training loss towards a constant, non-zero flood level, while the state-of-the-art iFlood regularizer (Xie et al., 2022) applies a constant, non-zero flood level to *each* training instance.

Training samples are, however, not uniformly difficult: some instances have more irreducible uncertainty than others (*i.e.* heteroskedastic noise), while some instances are simply easier to fit than others. It may not be beneficial to aggressively drive down the training loss for training samples that are outliers, noisy, or mislabeled. We explore this difference in the difficulty of training samples further in Section 3.1. To address this issue, we present Adaptive Flooding (AdaFlood), a novel flood regularizer that adapts the flood level of each training sample according to the difficulty of the sample (Section 3.2). We present theoretical support for AdaFlood in Section 3.4.

Like previous flood regularizers, AdaFlood is simple to implement and compatible with any optimizer. AdaFlood determines the appropriate flood level for each sample using an auxiliary network that is trained on a subset of training dataset. Adaptive flood levels need to be computed for each instance only once, in a pre-processing step prior to training the main network. The results of this pre-processing step are not specific to the main network, and so can be shared across multiple hyper-parameter tuning runs. Furthermore, we propose a significantly more efficient way to train an auxiliary model based on fine-tuning, which saves substantially in memory and computation, especially for overparameterized neural networks (Section 4.5).

Our experiments (Section 4) demonstrate that AdaFlood generally outperforms previous flood methods on a variety of tasks, including image and text classification, probability density estimation for asynchronous event sequences, and regression for tabular datasets. Models trained with AdaFlood are also more robust to noise (Section 4.3) and better-calibrated (Section 4.4) than those trained with other flood regularizers.

## 2 RELATED WORK

Regularization techniques have been broadly explored in the machine learning community to improve the generalization ability of neural networks. Regularizers augment or modify the training objective and are typically compatible with different model architectures, base loss functions, and optimizers. They can be used to achieve diverse purposes including reducing overfitting (Hanson & Pratt, 1988; Ioffe & Szegedy, 2015; Krogh & Hertz, 1991; Liang et al., 2021; Lim et al., 2022; Srivastava et al., 2014; Szegedy et al., 2016; Verma et al., 2019; Yuan et al., 2020; Zhang et al., 2018), addressing data imbalance (Cao et al., 2019; Gong et al., 2022), and compressing models (Ding et al., 2019; Li et al., 2020; Zhuang et al., 2020).

AdaFlood is a regularization technique for reducing overfitting. Commonly adopted techniques for reducing overfitting include weight decay (Hanson & Pratt, 1988; Krogh & Hertz, 1991), dropout (Liang et al., 2021; Srivastava et al., 2014), batch normalization (Ioffe & Szegedy, 2015), label smoothing (Szegedy et al., 2016; Yuan et al., 2020), and data augmentation (Lim et al., 2022; Verma et al., 2019; Zhang et al., 2018). Inspired by work on overparametrization and double descent (Belkin et al., 2019; Nakkiran et al., 2021), Ishida et al. (2020) proposed a technique that aims to prevent the training loss from reaching zero by maintaining a small constant value; they termed this "flooding" by analogy to keeping the bottom of a container flooded with water. Xie et al. (2022) investigated instability in flooding, as it can lead to different solutions inconsistent in their generalization abilities and predictions for individual data points. They proposed an individual flooding loss function, called iFlood, to suppress confidence on over-fit examples while better fitting under-fitted instances. In contrast to the original flood regularizer, which encourages the *overall* training loss towards a constant target, iFlood drives *each* training sample's loss towards some constant $b$.

AdaFlood instead uses an auxiliary model trained on a heldout dataset to assign an adaptive flood level to each training sample. Using a heldout dataset to condition the training of the primary model is a well-known strategy in machine learning, and is regularly seen in meta-learning (Bertinetto et al., 2919; Franceschi et al., 2018), batch or data selection (Fan et al., 2018; Mindermann et al., 2022), and neural architecture search (Liu et al., 2019; Wang et al., 2021), among other areas.

## 3 ADAPTIVE FLOODING

Adaptive Flooding (AdaFlood) is a general regularization method for training neural networks; it can accommodate any typical loss function and optimizer.

### 3.1 PROBLEM STATEMENT

**Background**   Given a labeled training dataset $\mathcal{D} = \{(\boldsymbol{x}_i, y_i)\}_{i=1}^{N}$, where $\boldsymbol{x}_i \in \mathcal{X}$ are data samples and $y_i \in \mathcal{Y}$ are labels, we train a neural network $f : \mathcal{X} \to \widehat{\mathcal{Y}}$ by minimizing a training loss $\ell : \mathcal{Y} \times \widehat{\mathcal{Y}} \to \mathbb{R}$. In supervised learning we usually have $\ell \geq 0$, but in settings such as density estimation it may be negative. While conventional training procedures attempt to minimize the average training loss, this can lead to overfitting on training samples.

The original flood regularizer (Ishida et al., 2020) defines a global flood level for the average training loss, attempting to reduce the "incentive" to overfit. Denote the average training loss by $\mathcal{L} = \frac{1}{B} \sum_{i=1}^{B} \ell(y_i, f(x_i))$, where $f(x_i)$ denotes the model prediction and $B$ is the size of a mini-batch. Instead of minimizing $\mathcal{L}$, Flood (Ishida et al., 2020) regularizes the training by minimizing

$$\mathcal{L}_{\text{Flood}} = |\mathcal{L} - b| + b, \tag{1}$$

where the hyperparameter $b$ is a fixed flood level. Individual Flood (iFlood) instead assigns a "local" flood level, trying to avoid instability observed with Flood (Xie et al., 2022):

$$\mathcal{L}_{\text{iFlood}} = \frac{1}{B} \sum_{i=1}^{B} \big( |\ell(y_i, f(\boldsymbol{x}_i)) - b| + b \big). \tag{2}$$

**Motivation**   Training samples are, however, not uniformly difficult: some are inherently easier to fit than others. Figure 1a shows the dispersion of difficulty on CIFAR10 and CIFAR100 with various

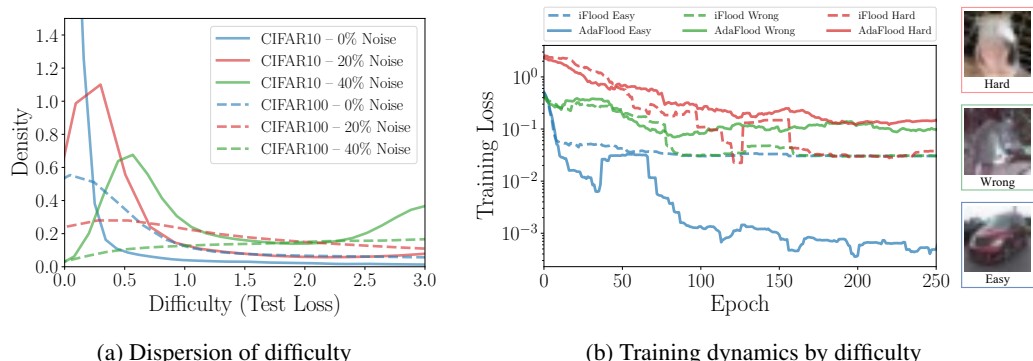

(a) Dispersion of difficulty          (b) Training dynamics by difficulty

Figure 1: (a) Illustration of how difficulties of examples are dispersed with and without label noise (where the relevant portion of examples have their label switched to a random other label). (b) Comparison of training dynamics on some examples between iFlood and AdaFlood. The "Hard" example is labeled *horse*, but models usually predict *cow*; the "Wrong" example is incorrectly labeled in the dataset as *cat* (there is no *rat* class).

levels of added label noise, as measured by the heldout cross-entropy loss from cross-validated models. Although difficulties on CIFAR10 without added noise are concentrated around difficulty $\leq 0.5$, as the noise increases, they vastly spread out. CIFAR100 has a wide spread in difficulty, even without noise. A constant flood level as used in iFlood may be reasonable for un-noised CIFAR10, but it seems less appropriate for CIFAR100 or noisy-label cases.

Moreover, it may not be beneficial to aggressively drive the training loss for training samples that are outliers, noisy, or mislabeled. In Figure 1b, we show training dynamics on an *easy*, *wrong*, and a *hard* example from the training set of CIFAR10. With iFlood, each example's loss converges to the pre-determined flood level (0.03); with AdaFlood, the *easy* example converges towards zero loss, while the *wrong* and *hard* examples maintain higher loss.

## 3.2 Proposed Method: AdaFlood

Differences in per-sample difficulty are the basis of many advances in efficient neural network training and inference, such as batch or data selection (Coleman et al., 2020; Fan et al., 2018; Mindermann et al., 2022) and dynamic neural networks (Li et al., 2021; Verelst & Tuytelaars, 2020). AdaFlood connects this observation to flooding. Intuitively, easy training samples (e.g. a correctly-labeled image of a *cat* in a typical pose) can be driven more aggressively to zero training loss without overfitting the model, while doing so for noisy, outlier, or incorrectly-labeled training samples may cause overfitting. These types of data points behave differently during training (Ren et al., 2022), and so should probably not be treated the same. AdaFlood does so by setting a sample-specific flood level in its objective:

$$\mathcal{L}_{\text{AdaFlood}} = \frac{1}{B} \sum_{i=1}^{B} \left( |\ell(y_i, f(\boldsymbol{x}_i)) - \theta_i| + \theta_i \right). \tag{3}$$

Here the sample-specific parameters $\theta_i$ should be set according to the individual sample's difficulty. AdaFlood estimates this quantity according to

$$\theta_i = \ell(y_i, \phi_\gamma(f^{\text{aux},i}(\boldsymbol{x}_i), y_i)), \tag{4}$$

where $f^{\text{aux},i}$ is an auxiliary model trained with cross-validation such that $\boldsymbol{x}_i$ is in its heldout set, and $\phi_\gamma(\cdot)$ is a "correction" function explained in a moment. Figure 2 illustrates the training process using (3), Appendix A gives further motivation, and Section 3.4 gives further theoretical support.

The flood targets $\theta_i$ are fixed over the course of training the main network $f$, and can be pre-computed for each training sample prior to the first epoch of training $f$. We typically use five-fold cross-validation as a reasonable trade-off between computational expense and good-enough models to estimate $\theta_i$, but see further discussion in Section 3.3. The cost of this pre-processing step can be further amortized over many training runs of the main network $f$ since different variations and configurations of $f$ can reuse the adaptive flood levels.

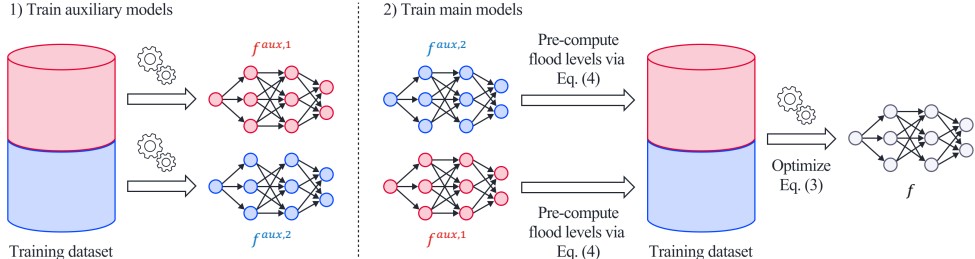

Figure 2: AdaFlood with data-efficiency trick for settings where training data is limited and acquiring additional data is impractical. In the first stage, we partition the training set into two halves and train two auxiliary networks $f^{\mathrm{aux},1}$ and $f^{\mathrm{aux},2}$: one on each half. In the second stage, we use each auxiliary network to set the adaptive flood level of training samples from the half it has not seen, via Eq. 4. The main network $f$ is then trained on the entire training set, minimizing the AdaFlood-regularized loss, Eq. 3. Note that the flood levels are fixed over the course of training $f$ and need to be pre-computed once only. The cost of pre-computation can be further amortized over many training runs of $f$ with different configurations.

---

**Algorithm 1** Training of Auxiliary Network(s) and AdaFlood

---

1: Train a single auxiliary network $f^{\mathrm{aux}}$ on the entire training set $\mathcal{D}$     ▷ Fine-tuning method only
2: **for** $\mathcal{D}^{\mathrm{aux},i}$ in $\{\mathcal{D}^{\mathrm{aux},i}\}_{i=1}^n$ **do**
3:      Train $f^{\mathrm{aux},i}$, either from scratch or by fine-tuning $f^{\mathrm{aux}}$, on $\mathcal{D} \setminus \mathcal{D}^{\mathrm{aux},i}$
4:      Save the adaptive flood level $\theta_i$ for each $\boldsymbol{x}_i \in \mathcal{D}^{\mathrm{aux},i}$ using $f^{\mathrm{aux},i}$ on $\boldsymbol{x} \in \mathcal{D}^{\mathrm{aux},i}$
5: **end for**
6: Train the main model $f$ using Equation (3) and adaptive flood levels $\theta$ computed above

---

**Correction function.** Unfortunately, the predictions from auxiliary models are not always correct even when trained on most of the training set – if they were, our model would be perfect already. In particular, the adaptive flood levels $\theta_i$ can be arbitrarily large for any difficult examples where the auxiliary model is incorrect; this could lead to strange behavior when we encourage the primary model $f$ to be very incorrect. We thus "correct" the predictions with the correction function $\phi_\gamma$, which mixes between the dataset's label and the heldout model's signal.

For **regression tasks**, the predictions $f(\boldsymbol{x}_i) \in \mathbb{R}$ should simply be close to the labels $y_i \in \mathbb{R}$. Here the correction function linearly interpolates the predictions and labels as,

$$\phi_\gamma(f^{\mathrm{aux}}(\boldsymbol{x}_i), y_i) = (1 - \gamma) f^{\mathrm{aux}}(\boldsymbol{x}_i) + \gamma y_i. \tag{5}$$

Here $\gamma = 0$ fully trusts the auxiliary models (no "correction"), while $\gamma = 1$ disables flooding.

For $K$-way **classification tasks**, $f(\boldsymbol{x}_i) \in \mathbb{R}^K$ is a vector of output probabilities (following a softmax layer), and the label is $y_i \in [K]$. Cross-entropy loss only considers the probability of true class $y_i$: $\ell(y, \hat{y}) = -\log(\hat{y}_{y_i})$. The $y_i$-th component of the correction function $\phi_\gamma(f^{\mathrm{aux}}(x_i), y_i)$ is then

$$\phi_\gamma(f^{\mathrm{aux},i}(\boldsymbol{x}_i), y_i)_{y_i} = (1 - \gamma) f^{\mathrm{aux},i}(\boldsymbol{x}_i)_{y_i} + \gamma. \tag{6}$$

Again, for $\gamma = 0$ there is no "correction," and for $\gamma = 1$ flooding is disabled, as $\theta_i = -\log 1 = 0$.

The hyperparameter $\gamma \in [0, 1]$ is perhaps simpler to interpret and search for than directly identifying a flood level as in Flood or iFlood; in those cases, the level is unbounded (in $[0, \infty)$ for supervised tasks and all of $\mathbb{R}$ for density estimation) and the choice is quite sensitive to the particular task.

### 3.3 Efficiently Training Auxiliary Networks

Although the losses from auxiliary networks can often be good measures for the difficulties of samples, this is only true when the number of folds $n$ is reasonably large; otherwise the training set of size about $\frac{n-1}{n}|\mathcal{D}|$ may be too much smaller than $\mathcal{D}$ for the model to have comparable performance. The computational cost scales roughly linearly with $n$, however, since we must train $n$ auxiliary

networks: if we do this in parallel it requires $n$ times the computational resources, or if we do it sequentially it takes $n$ times as long as training a single model.

To alleviate the computational overhead for training auxiliary networks, we sometimes instead approximate the process by fine-tuning a single auxiliary network. More specifically, we first train a single base model $f^{\mathrm{aux}}$ on the entire training set $\mathcal{D}$. We then train each of the $n$ auxiliary models by randomly re-initializing the last few layers, then re-training with the relevant fold held out. The process is illustrated in Figure 7 and Algorithm 1.

Although this means that $\boldsymbol{x}_i$ does slightly influence the final prediction $f^{\mathrm{aux},i}(\boldsymbol{x}_i)$ ("training on the test set"), it is worth remembering that we use $\theta_i$ only as a parameter in our model: $\boldsymbol{x}_i$ is in fact a training data point for the overall model $f$ being trained. This procedure is justified by recent understanding in the field that in typical settings, a single data point only loosely influence the early layers of a network. In highly over-parameterized settings (the "kernel regime") where neural tangent kernel theory is a good approximation to the training of $f^{\mathrm{aux}}$ (Jacot et al., 2018), re-initializing the last layer would completely remove the effect of $\boldsymbol{x}_i$ on the model. Even in more realistic settings, although the mechanism is not yet fully understood, last layer re-training seems to do an excellent job at retaining "core" features and removing "spurious" ones that are more specific to individual data points (Kirichenko et al., 2023; LaBonte et al., 2023).

For smaller models with fewer than a million parameters, we use 2- or 5-fold cross-validation, since training multiple auxiliary models is not much of a computational burden. For larger models such as ResNet18, however, we use the fine-tuning method. This substantially reduces training time, since each fine-tuning gradient step is less expensive and the models converge much faster given strong features from lower levels than they do starting from scratch; Section 4.5 gives a comparison.

To validate the quality of the flood levels from the fine-tuned auxiliary network, we compare them to the flood levels from $n = 50$ auxiliary models using ResNet18 (He et al., 2016) on CIFAR10 (Krizhevsky et al., 2009); with $n = 50$, each model is being trained on 98% of the full dataset, and thus should be a good approximation to the best that this kind of method can achieve. The Spearman rank correlation between the fine-tuned method and the full cross-validation is 0.63, a healthy indication that this method provides substantial signal for the "correct" $\theta_i$. Our experimental results also reinforce that this procedure chooses a reasonable set of parameters.

### 3.4 Theoretical Intuition

For a deeper understanding of AdaFlood's advantages, we now examine a somewhat stylized supervised learning setting: an overparameterized regime where the $\theta_i$ are nonetheless optimal.

**Proposition 1.** *Let $\mathcal{F}$ be a set of candidate models, and suppose there exists an optimal model $f^* \in \arg\min_{f \in \mathcal{F}} \mathbb{E}_{\boldsymbol{x},y}\ell(y, f(\boldsymbol{x}))$, where $\ell$ is a nonnegative loss function. Given a dataset $\mathcal{D} = \{(\boldsymbol{x}_i, y_i)\}_{i=1}^N$, let $\hat{f}$ denote a minimizer of the empirical loss $\mathcal{L}(f) = \frac{1}{N}\sum_{i=1}^N \ell(y_i, f(\boldsymbol{x}_i))$; suppose that, as in an overparameterized setting, $\mathcal{L}(\hat{f}) = 0$. Also, let $\bar{f}$ be a minimizer of the AdaFlood loss (3) using "perfect" flood levels $\theta_i = \ell(y_i, f^*(\boldsymbol{x}_i))$. Then we have that*

$$\mathcal{L}(\hat{f}) = 0 \leq \mathcal{L}(f^*) = \mathcal{L}(\bar{f}), \quad \mathcal{L}_{AdaFlood}(\hat{f}) = 2\mathcal{L}(f^*) \geq \mathcal{L}(f^*) = \mathcal{L}_{AdaFlood}(f^*) = \mathcal{L}_{AdaFlood}(\bar{f}).$$

We know that $\mathcal{L}(f^*)$ will be approximately the Bayes error, the irreducible distributional error achived by $f^*$; this holds for instance by the law of large numbers, since $f^*$ is independent of the random sample $\mathcal{D}$. Thus, if the Bayes error is nonzero and the $\theta_i$ are optimal, we can see that empirical risk minimization of overparametrized models will find $\hat{f}$, and disallow $f^*$; minimizing $\mathcal{L}_{\mathrm{AdaFlood}}$, on the other hand, will allow the solution $f^*$ and disallow the empirical risk minimizer $\hat{f}$.

*Proof.* With this choice of $\theta_i$, we have that

$$\mathcal{L}_{\mathrm{AdaFlood}}(f) = \frac{1}{N}\sum_{i=1}^N \Big( |\ell(y_i, f(\boldsymbol{x}_i)) - \ell(y_i, f^*(\boldsymbol{x}_i))| + \ell(y_i f^*(\boldsymbol{x}_i)) \Big).$$

Since the absolute value is nonnegative, we have that $\mathcal{L}_{\mathrm{AdaFlood}}(f) \geq \mathcal{L}(f^*)$ for any $f$, and that $\mathcal{L}_{\mathrm{AdaFlood}}(f^*) = \mathcal{L}(f^*)$; this establishes that $f^*$ minimizes $\mathcal{L}_{\mathrm{AdaFlood}}$, and that any minimizer $\bar{f}$ must

| NTPP | Method | Uber | | Reddit | | | Stack Overflow | | |
|---|---|---|---|---|---|---|---|---|---|
| | | RMSE | NLL | RMSE | NLL | ACC | RMSE | NLL | ACC |
| Intensity-free | Unrge. | 75.83 | 3.86 | 0.25 | 1.28 | 55.26 | 6.69 | 3.66 | 45.52 |
| | | (6.12) | (0.05) | (0.01) | (0.07) | (0.57) | (0.98) | (0.12) | (0.07) |
| | Flood | 64.34 | 4.01 | 0.25 | 1.17 | 57.46 | 4.12 | 3.46 | **45.76** |
| | | (3.85) | (0.02) | (0.01) | (0.06) | (0.84) | (0.23) | (0.03) | **(0.03)** |
| | iFlood | 67.07 | 3.97 | **0.23** | 1.11 | 56.59 | 4.12 | 3.46 | **45.76** |
| | | (3.12) | (0.06) | **(0.01)** | (0.12) | (0.92) | (0.23) | (0.03) | **(0.03)** |
| | AdaFlood | **59.69** | **3.75** | 0.26 | **1.09** | **59.02** | **3.26** | **3.45** | 45.67 |
| | | **(1.49)** | **(0.01)** | (0.02) | **(0.13)** | **(0.91)** | **(0.25)** | **(0.04)** | (0.03) |
| THP$^+$ | Unreg. | 71.01 | 3.73 | 0.28 | 0.82 | 58.63 | 1.46 | 2.82 | 46.24 |
| | | (6.12) | (0.05) | (0.01) | (0.07) | (0.57) | (0.98) | (0.12) | (0.07) |
| | Flood | 68.61 | 3.70 | 0.26 | 1.02 | 58.05 | 1.39 | 2.79 | 46.31 |
| | | (3.85) | (0.02) | (0.01) | (0.06) | (0.84) | (0.23) | (0.03) | (0.03) |
| | iFlood | 68.61 | 3.70 | **0.25** | 0.92 | 58.93 | 1.46 | 2.82 | 46.24 |
| | | (4.76) | (0.17) | (0.01) | (0.23) | (1.26) | (0.06) | (0.04) | (0.08) |
| | AdaFlood | **54.85** | **3.55** | **0.25** | **0.80** | **61.34** | **1.38** | **2.77** | **46.41** |
| | | **(1.49)** | **(0.01)** | **(0.02)** | **(0.13)** | **(0.91)** | **(0.25)** | **(0.04)** | **(0.03)** |

Table 1: Comparison of flooding methods on asynchronous event sequence datasets. The numbers are the means and standard errors (in parentheses) over three runs.

achieve $\ell(y_i, \bar{f}(\boldsymbol{x}_i)) = \theta_i$ for each $i$, so $\mathcal{L}(\bar{f}) = \mathcal{L}(f^*)$. Using that $\ell(y_i, \hat{f}(\boldsymbol{x}_i)) = 0$ for each $i$, as is necessary for $\ell \geq 0$ when $\mathcal{L}(\hat{f}) = 0$, shows $\mathcal{L}_{\text{AdaFlood}}(\hat{f}) = \frac{1}{N} \sum_{i=1}^{N} 2\theta_i = 2\mathcal{L}(f^*)$. $\qquad \square$

In settings where $\theta_i$ is not perfect (and we would not expect the auxiliary models to obtain *perfect* estimates of the loss) the comparison will still approximately hold. If $\theta_i$ consistently overestimates the $f^*$ loss, $f^*$ will still be preferred to $\hat{f}$: for instance, if $\theta_i = 2\ell(y_i, f^*(\boldsymbol{x}_i))$, then $\mathcal{L}_{\text{AdaFlood}}(\hat{f}) = 4\mathcal{L}(f^*) \geq 3\mathcal{L}(f^*) = \mathcal{L}_{\text{AdaFlood}}(f^*)$. On the other hand, if $\theta_i = \frac{1}{2}\ell(y_i, f^*(\boldsymbol{x}_i))$ – a not-unreasonable situation when using a correction function – then $\mathcal{L}_{\text{AdaFlood}}(\hat{f}) = \mathcal{L}(f^*) = \mathcal{L}_{\text{AdaFlood}}(f^*)$. When $\theta_i$ is random, the situation is more complex, but we can expect that noisy $\theta_i$ which somewhat overestimate the loss of $f^*$ will still prefer $f^*$ to $\hat{f}$.

# 4 EXPERIMENTS

We now demonstrate the effectiveness of AdaFlood on three tasks (probability density estimation, classification and regression) in four different domains (asynchronous event sequences, image, text and tabular). We compare flooding methods on asynchronous event time in Section 4.1 and image classification tasks in Section 4.2. We also demonstrate that AdaFlood is more robust to various noisy settings in Section 4.3, and that it yields better-calibrated models for image classification tasks in Section 4.4. We investigate the performance of the fine-tuning scheme in Section 4.5.

## 4.1 RESULTS ON ASYNCHRONOUS EVENT SEQUENCES

In this section, we compare flooding methods on asynchronous event sequence datasets of which goal is to estimate the probability distribution of the next event time given the previous event times. Each event may or may not have a class label. Asynchronous event sequences are often modeled as temporal point processes and terms are used interchangeably. Details are provided in Appendix B.

**Datasets** We use two popular benchmark datasets, Stack Overflow (predicting the times at which users receive badges) and Reddit (predicting posting times). Following Bae et al. (2023), we also benchmark our method on a dataset with stronger periodic patterns: Uber (predicting pick-up times). We split each training dataset into train (80%) and validation (20%) sets.

| Method | SVHN | | CIFAR10 | | CIFAR100 | |
|---|---|---|---|---|---|---|
| | w/o $L_2$ reg. | w/ $L_2$ reg. | w/o $L_2$ reg. | w/ $L_2$ reg. | w/o $L_2$ reg. | w/ $L_2$ reg. |
| Unreg. | $95.65 \pm 0.05$ | $96.07 \pm 0.01$ | $87.80 \pm 0.31$ | $90.35 \pm 0.21$ | $56.59 \pm 0.32$ | $61.49 \pm 0.16$ |
| Flood | $95.63 \pm 0.02$ | $96.13 \pm 0.02$ | $87.57 \pm 0.16$ | $90.09 \pm 0.20$ | $55.88 \pm 0.18$ | $60.96 \pm 0.03$ |
| iFlood | $95.63 \pm 0.08$ | $96.05 \pm 0.02$ | $87.96 \pm 0.07$ | $90.57 \pm 0.12$ | $56.32 \pm 0.05$ | $61.63 \pm 0.12$ |
| AdaFlood | $\mathbf{95.72 \pm 0.01}$ | $\mathbf{96.16 \pm 0.02}$ | $\mathbf{88.38 \pm 0.18}$ | $\mathbf{90.82 \pm 0.08}$ | $\mathbf{57.25 \pm 0.14}$ | $\mathbf{62.31 \pm 0.14}$ |

Table 2: Comparison of flooding methods on image classification datasets with and without $L_2$ regularization. The numbers are the means and standard errors over three runs.

Following the literature in temporal point processes (TPPs), we use two metrics to evaluate TPP models: *root mean squared error* (RMSE) and *negative log-likelihood* (NLL). While NLL can be misleadingly low if the probability density is mostly focused on the correct event time, RMSE is not a good metric if stochastic components of TPPs are ignored and a baseline is directly trained on the ground truth event times. Therefore, we train our TPP models on NLL and use RMSE at test time to ensure that we do not rely too heavily on RMSE scores and account for the stochastic nature of TPPs. When class labels for events are available, we also report the accuracy of class predictions.

**Implementation** For TPP models to predict the asynchronous event times, we employ Intensity-free models (Shchur et al., 2020) based on GRU (Chung et al., 2014), and Transformer Hawkes Processes (THP) (Zuo et al., 2020) based on Transformer (Vaswani et al., 2017). THP predicts intensities to compute log-likelihood and expected event times, but this approach can be computationally expensive due to the need to compute integrals, particularly double integrals to calculate the expected event times. To overcome this challenge while maintaining performance, we follow Bae et al. (2023) in using a mixture of log-normal distributions, proposed in Shchur et al. (2020), for the decoder; we call this THP$^+$. The optimal flood levels are selected via a grid search on $\{-50, -45, -40 \dots, 0, 5\} \cup \{-4, -3 \dots, 3, 4\}$ for Flood and iFlood, and optimal $\gamma$ on $\{0.0, 0.1 \dots, 0.9\}$ for AdaFlood using the validation set. We use five auxiliary models.

**Results** In order to evaluate the effectiveness of various regularization methods, we present the results of our experiments in Table 1 (showing means and standard errors from three runs). This is the first time we know of where flooding methods have been applied in this domain; we see that all flooding methods improve the generalization performance here, sometimes substantially. Further, AdaFlood significantly outperforms the other methods for most models on most datasets, suggesting that the instance-wise flooding level adaptation using auxiliary models is a particularly effective way to enhance the generalization capabilities of both TPP models.

## 4.2 RESULTS ON IMAGE CLASSIFICATION

**Datasets** We use SVHN (Netzer et al., 2011), CIFAR-10, and CIFAR 100 (Krizhevsky et al., 2009) as the benchmarks for image classification with random crop and horizontal flip as augmentation. Unlike Xie et al. (2022), we split each training dataset into train (80%) and validation (20%) sets for hyperparameter search; thus our numbers are generally somewhat worse than what they reported.

**Implementation** On the image classification datasets, following Ishida et al. (2020) and similar to Xie et al. (2022), we consider training ResNet18 (He et al., 2016) on the datasets with and without $L_2$ regularization (with a weight of $10^{-4}$). All methods are trained with SGD for 300 epochs, with early stopping. We use a multi-step learning rate scheduler with an initial learning rate of 0.1 and decay coefficient of 0.2, applied at every 60 epochs. The optimal flood levels are selected based on validation performance with a grid search on $\{0.01, 0.02 \dots, 0.1, 0.15, 0.2 \dots, 1.0\}$ for Flood and iFlood, and $\{0.05, 0.1 \dots, 0.95\}$ for AdaFlood. We use a single ResNet18 auxiliary network where its layer 3 and 4 are randomly initialized and fine-tuned on held-out sets with $n = 10$ splits.

**Results** The results are presented in Table 2. We report the means and standard errors of accuracies over three runs. We can observe that flooding methods, including AdaFlood, are not significantly better than the unregularized baseline on SVHN. However, AdaFlood noticeably improves the per-

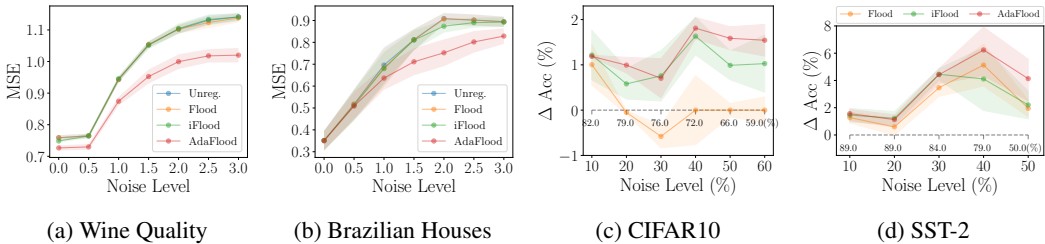

Figure 3: Comparison of flooding methods on tabular and image datasets with noise and bias.

formance over the other methods on harder datasets like CIFAR10 and CIFAR100, whereas iFlood is not obviously better than the baseline and Flood is worse than the baseline on CIFAR100.

### 4.3 NOISY LABELS

**Datasets** In addition to CIFAR10 for image classification, we also use the tabular datasets Brazilian Houses and Wine Quality from OpenML (Vanschoren et al., 2013), following Grinsztajn et al. (2022), for regression tasks. We further employ Stanford Sentiment Treebank (SST-2) for the text classification task, following Xie et al. (2022). Details of datasets are provided in Appendix B.

We report accuracy for classification tasks. For regression tasks, we report *mean squared error* (MSE) in the main body, as well as *mean absolute error* (MAE) and $R^2$ score in Figure 8.

**Implementation** We inject noise for both image and text classification by changing the label to a uniformly randomly selected wrong class, following Xie et al. (2022). More specifically, for $\alpha\%$ of the training data, we change the label to a uniformly random class other than the original label. For the regression tasks, we add errors sampled from a skewed normal distribution, with skewness parameter ranging from $0.0$ to $3.0$.

**Results** Figure 3 compares the flooding methods for noisy settings. We report the mean and standard error over three runs for CIFAR10, and five and seven runs for tabular datasets and SST-2, respectively. We provide $\Delta Acc\,(\%)$ for CIFAR10 and SST-2 compared to the unregularized model: that is, we plot the accuracy of each method minus the accuracy of the unregularized method, to display the gaps between methods more clearly. The mean accuracies of the unregularized method are displayed below the zero line.

- Wine Quality, Figure 3a: AdaFlood slightly outperforms the other methods at first, but the gap significantly increases as the noise level increases.
- Brazilian Houses, Figure 3b: There is no significant difference between the methods for small noise level, *e.g.* noise parameter $\leq 1.5$, but MSE for AdaFlood becomes significantly lower as the noise level increases.
- CIFAR10, Figure 3c: iFlood and AdaFlood significantly outperform Flood and unregularized. AdaFlood also outperforms iFlood when the noise level is high (*e.g.* $\geq 50\%$).
- SST-2, Figure 3d: Flooding methods significantly outperform the unregularized approach. AdaFlood is comparable to iFlood up to the noise level of $30\%$, but noticeably outperforms it as the noise level further increases.

Overall, AdaFlood is more robust to noise than the other flooding methods, since the model pays less attention to those samples with wrong or noisy labels.

### 4.4 CALIBRATION

**Datasets and implementation** Miscalibration – neural networks being over or under-confident – has been a well-known issue in deep learning. We thus evaluate the quality of calibration with different flooding methods on CIFAR100, as measured by the Expected Calibration Error (ECE) metric.

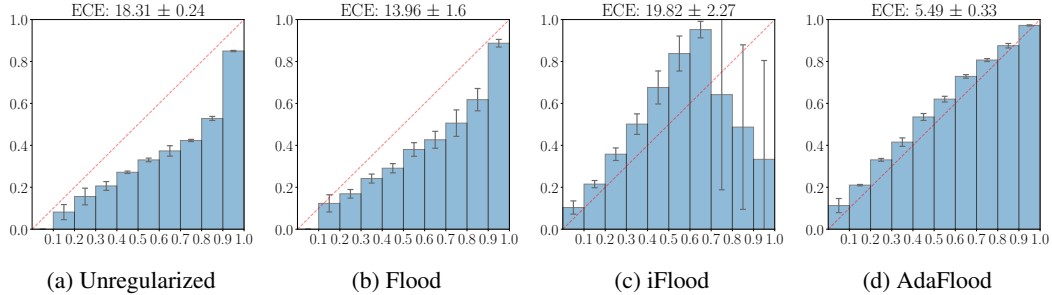

Figure 4: Calibration results of flooding methods with 10 bins on CIFAR100. The bars and errors are the means and standard errors over three runs, respectively.

(Figure 9 does the same for CIFAR10, but since model predictions are usually quite confident, this becomes difficult to measure.)

We use a ResNet18 with $L_2$ regularization with the optimal hyperparameters for the baseline and flooding methods. The optimal hyperparameter varies by seed for each run.

**Result** Figure 4 provides the calibration quality in ECE metric as well as a visualization over three runs, compared to perfect calibration (dotted red lines). We can observe that AdaFlood significantly improves the calibration, both in ECE and visually. Note that iFlood significantly miscalibrates at the bins corresponding to high probability *e.g.* bin $\geq 0.7$, compared to the other methods, and also has high standard errors. This behavior is expected, since iFlood encourages the model not to predict higher than a probability of $\exp(-b)$, where $b$ denotes the flood level used in iFlood.

### 4.5 ABLATION STUDY: FINE-TUNING VS. MULTIPLE AUXILIARIES

Figure 5 compares training of ten ResNet18 auxiliary networks (original proposal) to the single fine-tuned auxiliary network (efficient variant) in terms of wall-clock time for training using an NVIDIA GTX 1080 Ti GPU, and performance of the corresponding main model, on CIFAR10. Furthermore, we provide the training time and performance of an auxiliary network fine-tuned on different layers: $Layer3, 4 + FC$, $Layer3 + FC$, and $FC$, where $Layer3$ and $4$ are the 3rd, 4th layers in ResNet18 and $FC$ denotes the last fully connected layer.

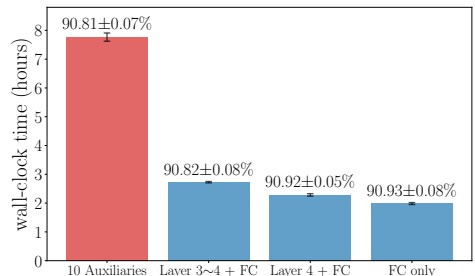

Figure 5: Comparison of aux. training

The result shows that training multiple auxiliary networks yields the same-quality model as fine-tuning, although training time is about $3 - 4$ times longer. There is also little difference in performance between different fine-tuning methods: it seems that fine-tuning only the FC layer is sufficient to forget the samples, with early-stopping regularizing well enough for similar generalization ability.

## 5 CONCLUSION

In this paper, we introduced the Adaptive Flooding (AdaFlood) regularizer, a novel regularization technique that adaptively regularizes a loss for each sample based on the difficulty of the sample. Each flood level is computed only once through an auxiliary training procedure with held-out splitting, which we can make more efficient by fine-tuning the last few layers on held-out sets. Experimental results on various domains and tasks: density estimation for asynchronous event sequences, image and text classification tasks as well as regression tasks on tabular datasets, with and without noise, demonstrated that our approach is more robustly applicable to a varied range of tasks including calibration.

**Reproducibility** For each experiment, we listed implementation details of the experiment such as model, optimizer, learning rate scheduler, regularization, and search space for hyperparameters. We also specify datasets we used for each experiment, and how they were split and augmented, along with the description of metrics. The code will be released in the final version.

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

## A  WHY WE CALCULATE $\theta$ USING HELD-OUT DATA

In Section 3.2, we estimate $\theta_i$ for each training sample using the output of an auxiliary network $f^{\mathrm{aux}}(x_i)$ that is trained on a held-out dataset. In fact, this adaptive flood level $\theta_i$ can be considered as the sample difficulty when training the main network. Hence, it is reasonable to consider existing difficulty measurements based on learning dynamics, like C-score (Jiang et al., 2021) or forgetting score (Maini et al., 2022). However, we find these methods are not robust when wrong labels exist in the training data, because the network will learn to remember the wrong label of $x_i$, and hence provide a low $\theta_i$ for the wrong sample, which is harmful to our method. That is why we propose to split the whole training set into $n$ parts and train $f^{\mathrm{aux}}(x_i)$ for $n$ times (each with different $n-1$ parts).

**Dataset and implementation**     To verify this, we conduct experiments on a toy Gaussian dataset, as illustrated in the first panel in Figure 6. Assume we have $N$ samples, each sample in 2-tuple $(x, y)$. To draw a sample, we first select the label $y = k$ following a uniform distribution over all $K$ classes. After that, we sample the input signal $x \mid (y = k) \sim \mathcal{N}(\mu_k, \sigma^2 I)$, where $\sigma$ is the noise level for all the samples. $\mu_k$ is the mean vector for all the samples in class $k$. Each $\mu_k$ is a 10-dim vector, in which each dimension is randomly selected from $\{-\delta_\mu, 0, \delta_\mu\}$. Such a process is similar to selecting 10 different features for each class. We consider 3 types of samples for each class: regular samples, the typical or easy samples in our training set, have a small $\sigma$; irregular samples have a larger $\sigma$; mislabeled samples have a small $\sigma$, but with a flipped label. We generate two datasets following this same procedure (call them datasets $A$ and $B$). The, we randomly initialize a 2-layer MLP with ReLU layers and train it on dataset $A$. At the end of every epoch, we record the loss of each sample in dataset $A$.

**Result**     The learning paths are illustrated in the second panel in Figure 6. The model is clearly able remember all the wrong labels, as all the curves converge to a small value. If we calculate $\theta_i$ in this way, all $\theta_i$ would have similar values. However, if we instead train the model using dataset $B$, which comes from the same distribution but is different from dataset $A$, the learning curves of samples in dataset $A$ will behave like the last panel in Figure 6. The mislabeled and some irregular samples can be clearly identified from the figure. Calculating $\theta_i$ in this way gives different samples more distinct flood values, which makes our method more robust to sample noise, as our experiments on various scenarios show.

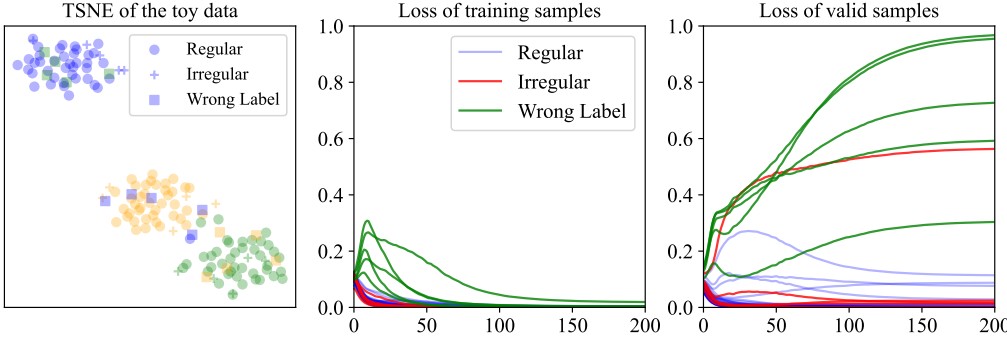

Figure 6: Left: the t-SNE (Van der Maaten & Hinton, 2008) of toy Gaussian example; middle: loss of different samples in the training set; right: loss of different samples in the validation set.

## B  DETAILS ABOUT DATASETS

**Stack Overflow**     It contains $6,633$ sequences with $480,414$ events where an event is the acquisition of badges received by users. The maximum number of sequence length is $736$ and the number of marks is $22$. The dataset is provided by Du et al. (2016); we use the first folder, following Shchur et al. (2020) and Bae et al. (2023).

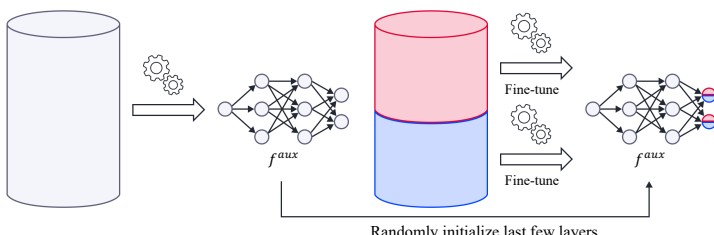

Figure 7: Efficient fine-tuning method for training a auxiliary network when held-out split is $n = 2$.

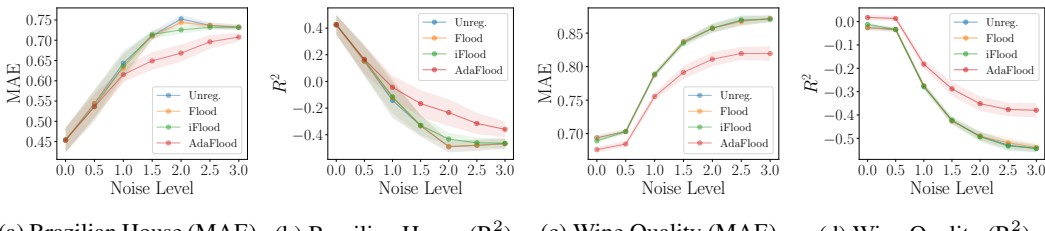

(a) Brazilian House (MAE)  (b) Brazilian House ($R^2$)  (c) Wine Quality (MAE)  (d) Wine Quality ($R^2$)

Figure 8: Additional results in various metrics on tabular datasets with noise and bias

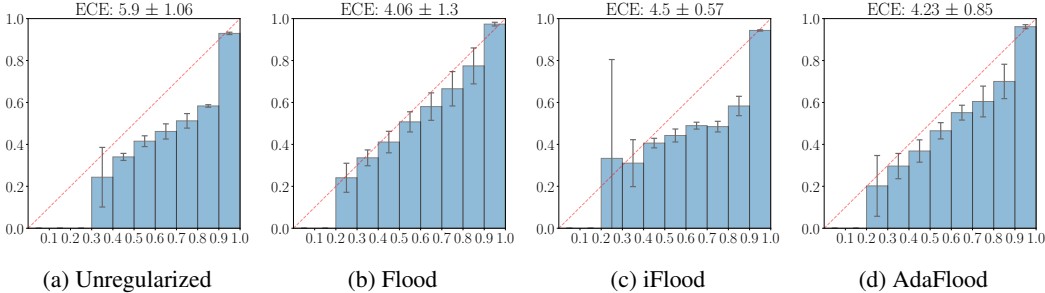

(a) Unregularized  (b) Flood  (c) iFlood  (d) AdaFlood

Figure 9: Calibration results of flooding methods with 10 bins on CIFAR10.

**Reddit**   It contains 10,000 sequences with 532,026 events where an event is posting in Reddit. The maximum number of sequence length is 736 and the number of marks is 22. Marks represent sub-reddit categories.

**Uber**   It contains 791 sequences with 701,579 events where an event is pick-up of customers. The maximum number of sequence length is 2,977 and there is no marks. It is processed and provided by Bae et al. (2023).

**Brazilian Houses**   It contains information of 10,962 houses to rent in Brazil in 2020 with 13 features. The target is the rent price for each house in Brazilian Real. According to OpenML (Vanschoren et al., 2013) where we obtained this dataset, since the data is web-scrapped, there are some values in the dataset that can be considered outliers.

**Wine Quality**   It contains 6,497 samples with 11 features and the quality of wine is numerically labeled as targets. This dataset is also obtained from OpenML (Vanschoren et al., 2013).

**SST-2**   The Stanford Sentiment Treebank (SST-2) is a dataset containing fully annotated parse trees, enabling a comprehensive exploration of how sentiment influences language composition. Comprising 11,855 individual sentences extracted from film reviews, this dataset underwent parsing using the Stanford parser, resulting in a collection of 215,154 distinct phrases.

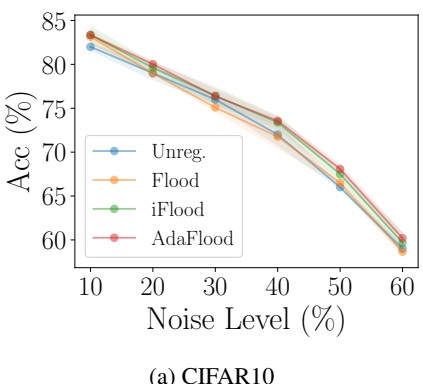
(a) CIFAR10

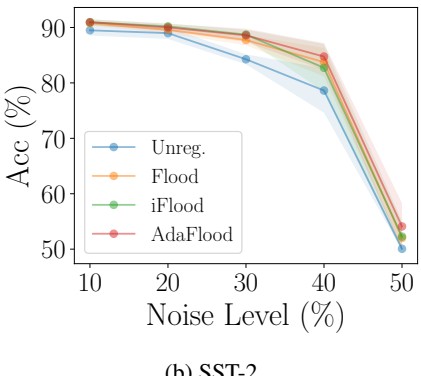
(b) SST-2

Figure 10: Visualization in Acc instead of $\Delta$Acc on CIFAR10 and SST-2 with varying noise levels.

| Method | Mislabled Sample Rate | | Method | Noise Level | | |
| --- | --- | --- | --- | --- | --- | --- |
| | 0% | 30% | | 0.0 | 1.5 | 3.0 |
| Unreg. | 81.00 (6.64) | 68.12 (17.23) | Unreg. | **0.2373 (0.3335)** | 0.3707 (-0.0409) | 0.3910 (-0.0978) |
| Flood | 81.18 (6.44) | 68.19 (17.71) | Flood | **0.2373 (0.3335)** | 0.3707 (-0.0409) | 0.3904 (-0.0980) |
| iFlood | 81.04 (6.86) | 68.24 (20.67) | iFlood | 0.2370 (0.3374) | 0.3652 (-0.0255) | 0.3902 (-0.0986) |
| AdaFlood | **81.79 (4.81)** | **69.22 (17.45)** | AdaFlood | 0.2369 (0.3348) | **0.3520 (0.0250)** | **0.3465 (0.0119)** |

Table 3: Comparison of flooding methods on ImageNet100 (Left) and NYC Taxi Tip (Right) datasets with and without label noise. The numbers in parentheses on ImageNet100 represents ECE metric whereas on NYC Taxi Tip, they are $R^2$ scores.

## C  ADDITIONAL RESULTS ON IMAGE CLASSIFICATION

**Datasets**   We use ImageNet100 (Tian et al., 2020) for image classification with random crop, horizontal flip, and color jitter as augmentation. We also add $30\%$ of label noise as done in Section 4.3.

**Implementation**   We train ResNet34 (He et al., 2016) on the dataset with $L_2$ regularization (with a weight of $0.0001$). All methods are trained for 200 epochs with early stopping using SGD. We use a multi-step learning rate scheduler with an initial learning rate of $0.1$ and decay coefficient of $0.5$, applied at every 25 epochs. The optimal flood levels are selected based on validation performance with a grid search on $\{0.01, 0.02..., 0.1, 0.15, 0.2..., 0.3\}$ for Flood and iFlood, and $\{0.05, 0.1..., 0.95\}$ for AdaFlood. We use a single ResNet34 auxiliary network where its last FC layer is randomly initialized and fine-tuned on held-out sets with $n = 10$ splits.

**Results**   The table below compares flooding methods on ImageNet100 dataset with and without $30\%$ of label noise. We report test accuracies along with expected calibration error (ECE) in parentheses. Although Flood and iFlood do not improve the performance over the unregularized model, AdaFlood improves the performance by about $0.80\%$ over the unregularized baseline. Given the size of the dataset, the gap is not marginal. This gap is even larger than that we observed in SVHN and CIFAR datasets Table 2. We conjecture it is because ImageNet contains more noisy samples. It is well-known that there are many ImageNet images containing multiple objects although the label says there is only one object.

The table below compares flooding methods on NYC Taxi Tip dataset (Grinsztajn et al., 2022) with and without noises. We report mean square error (MSE) and R2 score in parentheses. Note that R2 score is usually in between 0 and 1 but when predictions are bad, it can go below 0. From the table, we can observe that all flooding methods perform similar to the unregularized baseline when there is no noise. Although it continues for Flood and iFlood even under noisy settings, AdaFlood significantly outperforms the other methods when noise level is 1.5 and 3.0. In particular, while R2 scores of other methods go below 0, it does not happen with AdaFlood, which demonstrates the robustness of AdaFlood even for the large-scale dataset like NYC Taxi Tip.

# D   ADDITIONAL RESULTS ON TABULAR REGRESSION

**Datasets**   We use NYC Taxi Tip dataset from OpenML (Vanschoren et al., 2013), one of the largest tabular dataset used in Grinsztajn et al. (2022), for regression tasks. NYC Taxi Tip dataset contains $581,835$ rows and 9 features. As the name of the dataset implies the target variable is "tip amount". To increase the importance of other features, the creator of the dataset deliberately ignores "fare amount" or "trip distance".

**Implementation**   As with Section 4.3, we use a model tailored for tabular dataset proposed by (Grinsztajn et al., 2022) and add errors sampled from a skewed normal distribution, with skewness parameter ranging from $0.0$ to $3.0$.

**Results**   We report *mean squared error* (MSE) as well as $R^2$ scores in parantheses. As shown in Table 3 (right), AdaFlood is compatible to other methods but quickly improves (lower MSE and higher $R^2$ scores) the performance compared to the other methods as the noise level increases. It is consistent with the results we provided in Section 4.3.

# E   THEORETIC INTUITION FOR EFFICIENT TRAINING OF AUX. NETWORKS

In this section, we provide theoretic intuition for efficient training of auxiliary networks. Following (Lee et al., 2019), we approximate the predictions of a neural network $f$ on a test sample $x_i$

$$f(x_i) \approx \sum_{j=1}^{n} \alpha_j \text{eNTK}(x_i, x_j) \tag{7}$$

where eNTK stands for empirical neural tangent kernel (NTK) following (Mohamadi et al., 2023) and $\{x_j\}_{j=1}^n$ denotes data from a training set. Equation (7) says we can approximate a prediction on $x_i$ as an interpolation of $\text{eNTK}(x_i, \cdot)$ with some weights $\alpha$.

Suppose $f(x) = V\phi(x)$ where $\phi(x) \in \mathbb{R}^h$ denotes a feature from the penultimate layer and $V \in \mathbb{R}^{k \times h}$ denotes the weights of the last fully connected layer (k being the number of classes), consisting of $v_j \in \mathbb{R}^h$ for $j$-th row. If $v_{j,i}$, $i$-th entry of $v_j$, is from $N(0, \sigma^2)$, then Mohamadi et al. (2023) haven shown that,

$$\text{eNTK}_w(x_1, x_2)_{jj'} = v_j^T \text{eNTK}_{w \setminus V}^{\phi}(x_1, x_2)v_{j'} + \mathbb{1}(j = j')\phi(x_1)^T\phi(x_2) \tag{8}$$

where $w$ denotes a set of all the model parameters and $w \setminus V$ means a set of all the parameters except the last fully connected layer.

With this frame, we can approximate the predictions from the efficiently trained auxiliary network denoted as $f_{\text{tune}}$, as follows,

$$f_{\text{tune}}(x_i) \approx \sum_{j \neq i} \alpha_j^{\text{tune}} \left( v^T \text{eNTK}_{w \setminus V}^{\phi, \text{trained}}(x_j, x_i)v + \phi^{\text{trained}}(x_j)^T \phi^{\text{trained}}(x_i) \right). \tag{9}$$

Here, superscript "trained" means the model parameters are pre-trained on the whole training set.

On the other hand, the original "direct" training algorithm of auxiliary networks trains an auxiliary network from scratch, on the training set excluding the samples that we measure the difficulty on. Similarly, the predictions of a single auxiliary network denoted as $f_{\text{direct}}$ is approximated as,

$$f_{\text{direct}}(x_i) \approx \sum_{j \neq i} \alpha_j^{\text{direct}} \left( v^T \text{eNTK}_{w \setminus V}^{\phi, \text{untrained}}(x_j, x_i)v + \phi^{\text{untrained}}(x_j)^T \phi^{\text{untrained}}(x_i) \right) \tag{10}$$

where superscript "untrained" means the model parameters are randomly initialized.

In NTK regime where a neural network has infinite-width, the terms in the parentheses (the term except $\alpha$'s) are the same for Equation (9) and Equation (10). Therefore, the difficulty measures from the efficient training of a single auxiliary network and direct training of multiple auxiliary networks are equivalent in highly-overparameterized regime.

