# OpenReview forum: "AdaFlood: Adaptive Flood Regularization"
_ICLR.cc/2024/Conference — Submitted to ICLR 2024_

### Official Review · Reviewer_vFax · 2023-10-31

**Soundness:** 2 fair
**Presentation:** 3 good
**Contribution:** 2 fair
**Rating:** 5
**Confidence:** 3

**Summary:**

The author studies the flood regularization technique for relieving the overfitting of the deep neural network training and proposes an adaptive flood regularization technique, i.e., provides instance(sample)-wise adaptive weighting (flooding level) for calculating the flooding loss. This instance-wise weighting is determined by the auxiliary networks which are trained by subsets of the whole training dataset with five-fold cross-validation, and these auxiliary networks are fixed only to provide an estimation of the sample's difficulty, i.e., the loss between the auxiliary models' prediction and the input's ground-truth label. To mitigate the unavoidable imperfectness of the auxiliary model, the author also proposed a correction function to modify the prediction of the auxiliary model, i.e., linearly interpolates the predictions and the ground-truth labels. The author also provides ways to improve the training efficiency of the auxiliary networks by fine-tuning the last few layers of the auxiliary model such that they can use only one auxiliary model instead of multiple ones. The author also provides a theoretical intuition about the benefit of using adaptive flooding. Extensive experiments are conducted over text, images, asynchronous event sequences, and tabular data to demonstrate the effectiveness of the proposed method.

**Strengths:**

1. The paper is well-written, the idea of the present paper is reasonable, and the proposed technique is easy to follow.
2. The proposed method is simple to implement.
3. The benchmarking of the present paper is extensive.

**Weaknesses:**

1. The limited novelty of the proposed paper.

The idea of using adaptive weighting for model training is widely used in existing machine learning areas, the more relevant ones can be curriculum learning and uncertainty learning. Moreover, as mentioned by the author in Related Work, using a held-out dataset to condition the training of the primary model is a well-known strategy in machine learning, e.g., in meta-learning, batch or data selection, neural architecture search, etc. In the present paper, the author integrates the adaptive flooding level for flooding regularization, which is just a naive combination of the existing methods. Thus the novelty of the present paper is limited.

Although the author proposes a correction function to account for the imperfections of the prediction of the auxiliary model, the proposed correction function is very heuristic, and there is no theoretical analysis to reason about the soundness of this proposed function. The efficient training for the auxiliary networks also has a similar issue.

Overall, although the author provided a theoretical intuition for Equation (3), which is the training objective for adaptive flooding, the technical details to optimize this loss function proposed by the present paper are too heuristic and hard to be a principle and systematic approach.

2. The significance of the proposed method

Although the reviewer acknowledges the effort the author has made to validate their proposed method by the different modality of data, the data, and model chosen by the author is still not convincing. For example, though the SVHN, CIFAR10, and CIFAR100 are commonly used classification datasets, their complexity, scale, and reality are much less than ImageNet and more recent datasets like LAION. Although the review acknowledges that the author has stated in the last two lines of Page 7 that the result of the proposed method is not significant, this still gives us a feeling that the proposed method will not significantly improve the DNN training to avoid overfitting given so marginal improvement on Table 2. It is better for the author to consider a more realistic and decent scale of the dataset to conduct the experiment to show that the proposed simple method can achieve much better performance than state-of-the-art (SOTA), otherwise, it is hard to demonstrate the significance of the proposed method.

3. The efficient training strategy is mainly randomly re-initializing the last few layers of the model and then re-training with the relevant fold held out. How to define the term "few"? What's the principle here to choose those layers? The reviewer noticed that the author said ResNet-18 is a large model and used it in the experiment. Can the proposed method still work effectively when we use ViT for image classification and Transformer for text modality? Will this efficient training strategy still work?

4. The author states in the third paragraph of page 5 that:

"This procedure is justified by recent understanding in the field that in typical settings, a single data point only loosely influence
the early layers of a network."

The author should provide explicit references to support this claim, e.g. what recent understanding does the author refer to? How do they relate to the proposed method?

Minor:

1. The `Unreg.` in Figure 3 can not be seen. The author may consider another format of the line for better visualization.

**Questions:**

Please refer to the weakness section for the questions.

---

> ### Author Response · Authors · 2023-11-20
> **Response to Reviewer vFax (1/3)**
>
> Thank you for your comments. Please see the common response above as well as our responses below.
>
>
> > The limited novelty of the proposed paper. The idea of using adaptive weighting for model training is widely used in existing machine learning areas, the more relevant ones can be curriculum learning and uncertainty learning. using a held-out dataset to condition the training of the primary model is a well-known strategy in machine learning, e.g., in meta-learning, batch or data selection, neural architecture search, etc.
>
> Indeed, adaptive weighting and held-out datasets are each commonly used techniques in machine learning, as we mentioned in the paper. We are not aware, however, of any time they have been used together for something resembling the difficulty-aware regularization setting of our work. The vast majority of research is based on applying new combinations of existing tools in new settings; our paper is no different in this regard. We think this is a strength, and not a weakness: our method is built on well-tested tools that have individually proven effective in diverse fields, suggesting that our proposal is built on a solid foundation, and with the ability to share techniques and understanding of those tools across application areas.
>
> If we missed any closely related work in those areas, however, we are eager to compare them to our approach and cite them.
>
>
>
> > Although the author proposes a correction function to account for the imperfections of the prediction of the auxiliary model, the proposed correction function is very heuristic, and there is no theoretical analysis to reason about the soundness of this proposed function. The efficient training for the auxiliary networks also has a similar issue.
>
> We agree that the correction function is heuristic, but it is a very natural and intuitive way of correcting the predictions from auxiliary networks. It is very easy to implement, and it performs well (as we demonstrated in the multiple experiments).
>
> As the reviewer pointed out, it would be indeed better if we had solid theoretical background for every component, but in practice this is extremely rare, especially for deep learning. The influence of even the most basic regularizer, weight decay, is still a subject of very active research among deep learning theorists (e.g. [1] from earlier this year). We think that our support for AdaFlood in Proposition 1 establishes a strong theoretical baseline, and the discussion in the following paragraph gives sufficient initial intuition about the behavior of the correction function.
>
> For the efficient training of auxiliary networks, we expanded on the motivation in highly-overparameterized regimes in Appendix E of the revised paper. In summary, in the regime where networks are well-explained by their neural tangent kernels (NTKs), held-out predictions by an auxiliary network trained from scratch perfectly agree with predictions from a network trained according to our efficient algorithm. This is because the empirical NTK of the penultimate layer is identical for the two networks. Please refer to Appendix E for further details.
>
> [1] Joseph Shenouda, Rahul Parhi, Kangwook Lee, and Robert D. Nowak. Vector-Valued Variation Spaces and Width Bounds for DNNs: Insights on Weight Decay Regularization. arXiv:2305.16534.
>
>
>
> > The significance of the proposed method. Though the SVHN, CIFAR10, and CIFAR100 are commonly used classification datasets, their complexity, scale, and reality are much less than ImageNet and more recent datasets like LAION.
>
> We shared some results on larger datasets in the common response above, showing that AdaFlood still generally outperforms the other flooding methods (with even bigger gains in noisy-label settings). We are currently running experiments on the full ImageNet, and will add them to the revised paper once they are complete.
>
> We’d like to emphasize that, while we haven’t shown results on the largest image classification datasets, we have demonstrated the versatility of our method across a variety of tasks including density estimation, regression, and classification, on settings including asynchronous time sequences, images, tabular, and text data.

---

> ### Author Response · Authors · 2023-11-20
> **Response to Reviewer vFax (2/3)**
>
> > Although the author has stated […] the result of the proposed method is not significant, this still gives us a feeling that the proposed method will not significantly improve the DNN training to avoid overfitting given so marginal improvement on Table 2.
>
> We first want to highlight that in the last two lines of page 7, we were not saying that our method’s overall results were not significant – if that were the case, we wouldn’t have submitted the paper. Instead, we were saying that on _SVHN in particular_, AdaFlood performs about the same as previous flooding methods. (Performance is probably basically saturated for this class of model on SVHN.) AdaFlood does give nontrivial improvements in accuracy and calibration on CIFAR10 and CIFAR100. In the results in the common response above, we also see even larger improvements on ImageNet100 (probably because its labels are inherently noisier). Notice in particular that the improvement of AdaFlood over iFlood is larger than the improvement of iFlood over Flood, or of Flood over baseline models.
>
> Moreover, it is far more robust to noise on these datasets than competitor methods. Outside of image classification, it also sees improvements in both original and noisy versions of many datasets across asynchronous time sequence density estimation, text classification, tabular regression.
>
>
> > The efficient training strategy is […] How to define the term "few"? What's the principle here to choose those layers? Can the proposed method still work effectively when we use ViT for image classification and Transformer for text modality? Will this efficient training strategy still work?
>
> Recall that the purpose of having auxiliary networks is to measure difficulty of each sample. Here, we avoid a model seeing the sample that we measure difficulty on, in training time. With this motivation, random initialization of last few layers of efficient training of auxiliary networks is to forget information about the samples we measure difficulties on.
>
> According to the conditions provided by [1] for forgetting, randomly initializing only the last layer satisfies these conditions; initializing more layers, of course, does as well. Also, in [2] as the title implies, re-training only the last layer is sufficient to remove spurious correlations that are more specific to individual data points. These results suggest that re-initializing at least one layer should be sufficient.
>
> Then, the question is how different predictions will be depending on the number of layers we randomly initialize. [3] provides a user guide for fine-tuning, depending on the gap between pre-training and adaptation steps. According to [3], our fine-tuning falls in the category where the required feature update is tiny (since we fine-tune on a subset of the training set), which means whether we initialize only the last layer or last few layers, it won’t change the features too much at convergence.
>
> In addition to this guidance from prior work, we explore the question of how many layers to reinitialize in Section 4.5. We can see in those results that the performance of AdaFlood stays essentially the same (< 0.11%) if we reinitialize only the last layer, layer 4 plus the last, or layers 3 and 4 plus the last. Thus, reinitializing only the last layer gives the fastest method at essentially the same performance.
>
> Based on this, we agree that it will be clearer if we are more specific when presenting the method, and so changed the description in the revised paper from “the last few layers” to “the last layer.”
>
> As suggested, we also verified that the fine-tuning variant works for ViT. We took the ViT implementation from a publicly available repo (https://github.com/kentaroy47/vision-transformers-cifar10). To train multiple auxiliary networks, we split CIFAR10 into 20 folds, and trained 20 ViT models for $500$ epochs each as described in Sec 3.2. For the fine-tuning variant, we train a ViT model on the full CIFAR10, and then fine-tuned on 20 folds as described in Section 3.3. After collecting losses for $50,000$ training samples from both multiple and fine-tuned versions, we obtained a Spearman (rank) correlation between these predictions of 0.61, essentially the same as the correlation observed for ResNets. Thus, the efficient variant provides substantial signal to the “correct” adaptive flood levels with ViT as well.
>
> Lastly, we want to emphasize that the efficient training of auxiliary networks saves computational cost in computing the adaptive flood levels, but it is not a “must-have” component for the proposed AdaFlood method.
>
> [1] Hattie Zhou, Ankit Vani, Hugo Larochelle, Aaron Courville. “Fortuitous forgetting in connectionist networks”. ICLR 2022.
>
> [2] Polina Kirichenko, Pavel Izmailov, Andrew Gordon Wilson. “Last layer re-training is sufficientfor robustness to spurious correlations”. ICLR, 2023.
>
> [3] Yi Ren, Shangmin Guo, Wonho Bae, Danica J Sutherland. “How to Prepare Your Task Head for Finetuning”. ICLR 2023.

---

> ### Author Response · Authors · 2023-11-20
> **Response to Reviewer vFax (3/3)**
>
> > The author states in the third paragraph of page 5 that: "This procedure is justified by recent understanding in the field that in typical settings, a single data point only loosely influence the early layers of a network." The author should provide explicit references to support this claim, e.g. what recent understanding does the author refer to? How do they relate to the proposed method?
>
>
> Followed by the sentence the reviewer mentioned, we explained that “In highly over-parameterized settings (the “kernel regime”) where neural tangent kernel (NTK) theory is a good approximation to the training of $f^{aux}$ [1], re-initializing the last layer would completely remove the effect of $x_i$ on the model.” As mentioned for a previous question, we have added a more thorough discussion of this point to Appendix E.
>
> For further intuition about loose influence on the early layers, here is our reasoning:
>
> According to [1], in the NTK regime where the width of a neural network is infinite, and loss function is the least-square loss, the prediction from the neural network and ridgeless kernel regression using NTK are equal in expectation. In other words, the prediction of a neural network $f$ on a test sample $\tilde{x}$, say $f(\tilde{x})$, is governed by a fixed kernel called NTK. For ridgeless kernel regression, adding data points does not change the kernel function but it does change the prediction by adding a row and column to the kernel matrix. Conversely, if we remove $x_i$ from the training set, $x_i$ will not affect on inference at all.
>
> Even if a neural network does not have infinite-width, it is shown that overparameterized neural networks can converge linearly to zero training loss with their parameters hardly change [2] i.e., the change in the Jacobian matrix is very small. Hence, in this regime, a single data point only loosely influence training.
>
> In addition, [3] states that “…the gradient tends to get smaller as we move backward through the hidden layers. This means that neurons in the earlier layers learn much more slowly than neurons in later layers.” Combined with the previous arguments, we hope that our statement for “a single data point only loosely influence the early layers of a network” makes sense now.
>
> The paragraph where the we stated “a single data point only loosely influence the early layers of a network” was about justifying that re-initializing the last (few) layers of a pre-trained neural network is enough to forget about a data point, and so, it is reasonable to compute the difficulty on the point with a fine-tuned model. In the paragraph, we divided the parameter regimes as,
>
> - NTK regime — Re-initializing the last layer would completely remove the effect of $x_i$.
> - Non-NTK (more realistic) regime — Last layer re-training retains “core” features and removes “spurious” ones that are more specific to individual data points [4]
>
> If the reviewer thinks it is still unclear, please let us know. We are eager to discuss further and make the paper clearer.
>
>
>  [1] Arthur Jacot, Franck Gabriel, and Clement Hongler. “Neural tangent kernel: Convergence and generalization in neural networks”, NeurIPS 2018.
>
>  [2]Chizat, Lenaic and Oyallon, Edouard and Bach, Francis. “On lazy training in differentiable programming”, NeurIPS 2019
>
> [3] Michael A. Nielsen, "Neural Networks and Deep Learning", Determination Press, 2015
>
> [4] Polina Kirichenko, Pavel Izmailov, and Andrew Gordon Wilson. “Last layer re-training is sufficientfor robustness to spurious correlations”, ICLR, 2023.
>
>
>
> > The `Unreg.`  in Figure 3 can not be seen. The author may consider another format of the line for better visualization.
>
> As described in the Results paragraph in Section 4.3, we visualized $\Delta Acc$, defined as the improvement in accuracy of a flood method over the accuracy of `Unreg.`  We also provided the baseline `Unreg.` accuracy in numbers, below the $\Delta Acc$ lines. Visualizing in this way is helpful since the change in inherent difficulty between noise levels is much larger than the change in performance of different methods, and a plain accuracy plot (now shown as Figure 10 in the appendix) is overwhelmed by the difference in noise levels. We will consider other visualization formats, and would be happy to hear any suggestions that would be clearer while still making the difference between methods visible.

---

### Official Review · Reviewer_o6wC · 2023-10-31

**Soundness:** 3 good
**Presentation:** 3 good
**Contribution:** 2 fair
**Rating:** 3
**Confidence:** 4

**Summary:**

The paper presents a novel regularization technique dubbed AdaFlood, casting a spotlight on individual training sample difficulty by adaptively modulating the flood level. It parades its versatility and prowess across a spectrum of datasets and tasks like density estimation, classification, and regression, where it triumphs in generalization, robustness to noise, and calibration facets. A set of experimental validations heightens the paper, echoing the efficacy of AdaFlood in comparison to existing flooding methods.

**Strengths:**

(1) AdaFlood emerges as a trailblazing regularization strategy that sensitively tunes the flood level according to the intrinsic difficulty of each training sample.

(2) The experiments through the paper describe the AdaFlood’s versatility and superiority across a diverse array of tasks and datasets.

(3) Clarity and structural conformity characterize the paper’s presentation, weaving a coherent narrative that fluently elucidates the mechanics and nuances of AdaFlood.

**Weaknesses:**

(1) The method roots in a multi-view assumption, relying on auxiliary networks as different views to tailor their fitness. However, this assumption’s fragility surfaces in scenarios where auxiliary networks falter, casting doubts over the accurate estimation of flood levels.

(2) AdaFlood’s embrace of instance-wise reweighting and difficulty estimation unfolds as a labor-intensive odyssey, and its relevance dims when meeting vast training sets. The rigidly uniform and offline nature of difficulty estimation may misalign with the dynamic ebb and flow of instance contributions across various training stages.

(3) A confined exploration limited to toy datasets raises eyebrows, leaving unanswered questions regarding AdaFlood’s effectiveness when unleashed on larger, real-world, and more challenging benchmarks.

(4) The paper's theatrical stage seems restricted to closed-set supervised classification, leaving realms like self-supervised learning, like contrastive learning in vision and auto-regression in NLP, unnoticed.

(5) The improvement seems to be marginal compared with the baselines in both visual and NLP tasks, according to Tab. 1 and Tab. 2.

**Questions:**

N/A

---

> ### Author Response · Authors · 2023-11-20
> **Response to Reviewer o6wC (1/2)**
>
> Thank you for your comments. Please see the common response above as well as our responses below.
>
>
> > The method roots in a multi-view assumption, relying on auxiliary networks as different views to tailor their fitness. However, this assumption’s fragility surfaces in scenarios where auxiliary networks falter, casting doubts over the accurate estimation of flood levels.
>
> We first want to emphasize that we do not use a “multi-view assumption” in the sense typically used in the field. We use multiple auxiliary networks, but each one is used for a subset of the data, and “sees” the exact same features of the data – unlike multi-view learning, which uses multiple feature sets for the same underlying data points.
>
> The motivation of using multiple auxiliary models is simply that model’s training error on a particular data point is unlikely to be a good estimate of that data point’s inherent difficulty; instead, we use a cross-validation scheme to evaluate the held-out error for each data point.
>
> As to when “auxiliary networks falter”, we agree that utilizing auxiliary networks may harm training if auxiliary networks do not provide reasonably good predictions. Our correction function, introduced in Section 3.2, helps to mitigate this effect. As a reminder, the auxiliary network is not required to provide especially accurate predictions, but just to provide a useful-enough estimate of the difference in difficulty between samples to serve as an effective regularizer.
>
>
>
>
> >  AdaFlood’s embrace of instance-wise reweighting and difficulty estimation unfolds as a labor-intensive odyssey, and its relevance dims when meeting vast training sets.
>
>
> It is not easy to compute difficulty of samples, which is why there have been multiple works along that line such as C-score [1]. The challenge of computing difficulty lies on balance between learning good features and not seeing the samples we measure the difficult on. Due to this nature, the computation proportionally increases as the number of samples increases. It is, however, still meaningful to compute difficulties because it is a basis of many advances in efficient neural network training and inference [2, 3, 4]. Similarly, as we specified in Section 3.2, the cost of pre-computation can be further amortized over many training runs with different configurations or settings. We will release our measure of difficulties on image classification benchmark datasets to the public in the final version.
>
> More importantly, we proposed an efficient variant of computing difficulty. Using this efficient fine-tuning variant, it is not necessary to train multiple neural networks. In Section 3.3, we demonstrated that its difficulty measure is highly correlated to that from multiple auxiliary networks i.e., their rank correlation is 0.63. Furthermore, in Section 4.5, we demonstrated that this efficient variant is orders of magnitude faster than the original version, without significantly harming downstream performance.
>
> [1] Ziheng Jiang, Chiyuan Zhang, Kunal Talwar, and Michael C Mozer. “Characterizing structural regularities of labeled data in overparameterized models”. ICML 2021.
>
> [2] Cody Coleman, Christopher Yeh, Stephen Mussmann, Baharan Mirzasoleiman, Peter Bailis, Percy Liang, Jure Leskovec, and Matei Zaharia. “Selection via proxy: Efficient data selection for deep learning”. ICLR, 2020.
>
> [3] Yang Fan, Fei Tian, Tao Qin, Xiang-Yang Li, and Tie-Yan Liu. “Learning to teach”. ICLR, 2018.
>
> [4] Changlin Li, GuangrunWang, BingWang, Xiaodan Liang, Zhihui Li, and Xiaojun Chang. “Dynamic slimmable network”. In CVPR, 2021.

---

> ### Author Response · Authors · 2023-11-20
> **Response to Reviewer o6wC (2/2)**
>
> > The rigidly uniform and offline nature of difficulty estimation may misalign with the dynamic ebb and flow of instance contributions across various training stages.
>
> Our assumption is that there is a single “true” difficulty associated with each sample, i.e. its inherent noise, or the loss the Bayes-optimal predictor would find on that point. This is a standard assumption made in more realistic theoretical models from machine learning theory and in statistics, corresponding to heteroskedastic noise levels. A few recent works trying to measure this inherent difficulty are [1, 2].
>
> To our knowledge, no work in this area has considered dynamic notions of difficulty, which presumably would account for points which initially seem like noisy samples but eventually (once good-enough features are learned) are clear. Although this surely occurs over the training process, this behavior will be inherently tied to a model’s particular learning path, and so any such estimate will likely be far more tightly coupled to a particular model’s training process. We have a hard time picturing how to usefully estimate this in a fairly general setting, and think the gains may be limited compared to the effects from modeling heteroskedastic noise.
>
> [1] Ziheng Jiang, Chiyuan Zhang, Kunal Talwar, Michael C Mozer. “Characterizing structural regularities of labeled data in overparameterized models”. ICML 2021.
>
> [2] Pratyush Maini, Saurabh Garg, Zachary Lipton, J Zico Kolter. “Characterizing datapoints via
> second-split forgetting”. NeurIPS 2022.
>
>
>
> > A confined exploration limited to toy datasets raises eyebrows, leaving unanswered questions regarding AdaFlood’s effectiveness when unleashed on larger, real-world, and more challenging benchmarks.
>
> We disagree with characterizing the datasets used in this submission as “toy datasets,” and many of them are derived from real-world problems. For example, the Stack Overflow dataset contains 6,633 sequences with 480,414 events where an event is the acquisition of badges received by users in Stack Overflow; the Reddit dataset contains 10,000 sequences with 532,026 events where an event is posting in Reddit. Perhaps these days SVHN could be considered nearly a toy dataset, but there are still many papers in machine learning and computer vision with results focusing on CIFAR-10 and -100.
>
> As requested, however, we ran additional experiments on larger-scale datasets, with results described in the common response. Overall, we see that AdaFlood still helps over unregularized and other flooding-regularized models, with even larger improvements than before in the noisy-label setting.
>
>
>
> > The paper's theatrical stage seems restricted to closed-set supervised classification, leaving realms like self-supervised learning, like contrastive learning in vision and auto-regression in NLP, unnoticed.
>
> Our experiments have been conducted on many domains with several tasks: density estimation for asynchronous time sequences, regression for tabular datasets, classification for image and text datasets. Density estimation and regression are very different from classification tasks. Therefore, it is hard to agree that our method is restricted to closed-set supervised classification.
>
> We agree that we have not considered contrastive learning; it’s hard to see how to apply flooding-type regularizers in that setting. Autoregressive NLP tasks, however, are also a form of density estimation, and have characteristics of both the asynchronous time sequence datasets and supervised classification settings that we consider.
>
>
>
> > The improvement seems to be marginal compared with the baselines in both visual and NLP tasks, according to Tab. 1 and Tab. 2.
>
> We would like to correct that Table 1 is not for NLP tasks at all; it covers density estimation on asynchronous time sequences. More importantly, in Table 1, there are multiple measures where AdaFlood significantly outperforms the baselines: on the Uber dataset, the improvement over iFlood $7.38$ and $13.76$ in RMSE (standard error is around $1.5$), NLL: $0.22$ and $0.15$ (standard error is around $0.01$) with Intensity-free and THP+ models, respectively.
>
> For Table 2, as we specified, AdaFlood is not significantly better than SVHN but there is noticeable improvement on CIFAR10 and 100 (differences outside of standard errors). In addition, AdaFlood performs significantly better in calibration and in noisy settings.
>
> We provided further experiments on image classification datasets and tabular data in the previous response above. For those datasets, ours show more significant improvement over the baselines because the new datasets have more implicit noise compared to SVHN or CIFARs.
>
> We would also like to highlight, with regards to the claim that our improvements are marginal, that the improvement of AdaFlood over iFlood is generally speaking larger than the improvement of iFlood over Flood, and of Flood over baselines.

---

> > ### Comment · Reviewer_o6wC · 2023-11-23
> > **After Rebuttal**
> >
> > Thanks to the hard work of authors during response. But I cannot change the current rate due to the unconvincing arguments. Such instance-wise hardness reweighting by auxiliary networks looks like a multi-view-based semi/noisy-supervised learning. It is not an essentially innovated technique and hard to be applied for large-scale ML system with dynamic data flows.

---

> > > ### Author Response · Authors · 2023-11-23
> > > **Response to Reviewer o6wC**
> > >
> > > Dear Reviewer o6wC,
> > >
> > > Thank you for your valuable feedback. As you mentioned, the utilization of instance-wise hardness re-weighting with auxiliary networks may not be considered a novel technique. However, we would like to highlight that we are not aware of any instances where it has been applied in a manner resembling the difficulty-aware regularization setting presented in our work.
> > >
> > > The vast majority of research is based on applying new combinations of existing tools in new settings; our paper is no different in this regard. We believe this is a strength rather than a weakness of our paper. Our method is constructed on well-established tools that have demonstrated effectiveness across diverse fields, suggesting the robust foundation of our proposal, and the potential for sharing techniques and understanding across various application areas.
> > >
> > > Regarding the reviewer's mention of the technique's use in multi-view-based semi or noisy-supervised learning, we would like to explore these works further and compare them with our approach. Could you kindly provide a list of such works for our reference?
> > >
> > > Lastly, we'd like to ask for clarification on the reviewer's claim that "It is not an essentially innovated technique and hard to be applied for large-scale ML system with dynamic data flows." Does this imply that the technique employed in semi and noisy-supervised learning may not be scalable to large datasets? If there is supporting evidence for this claim, we would greatly appreciate getting additional insights for this aspect.
> > > Your additional input would be valuable in helping us better understand and address any concerns associated with this claim.
> > >
> > > Thank you for your time.

---

### Official Review · Reviewer_R2kv · 2023-11-01

**Soundness:** 2 fair
**Presentation:** 3 good
**Contribution:** 2 fair
**Rating:** 5
**Confidence:** 3

**Summary:**

This paper proposes a new strategy to define the optimization target. It assumes different samples have different difficulty. The experiments are conducted on various datasets, including CIFAR for image and Reddit, Uber, Stack Overflow for asynchronous event sequence.

**Strengths:**

-The proposed method is reasonable. Different samples has different difficulty. It is reasonable to use different loss.
-The paper is well-written and easy to follow.

**Weaknesses:**

-This paper lacks experiments on large-scale dataset, such as ImageNet. CIFAR and SVHN are small datasets, which are very different from the real sense.

-Does the adaptive strategy bring extra training cost? More detailed discussion is required.

**Questions:**

See the weakness.

---

> ### Author Response · Authors · 2023-11-20
> **Response to Reviewer R2kv**
>
> Thank you for your comments. Please see the common response above as well as our responses below.
>
>
> > This paper lacks experiments on large-scale dataset, such as ImageNet. CIFAR and SVHN are small datasets, which are very different from the real sense.
>
> Thank you for the suggestion. We shared some results larger datasets in the common response above, showing that AdaFlood still generally outperforms the other flooding methods (with even bigger gains in noisy-label settings). We are currently running experiments on the full ImageNet, and will add them to the revised paper once they are complete.
>
> We’d like to emphasize that, while we haven’t shown results on the largest image classification datasets, we have demonstrated the versatility of our method across a variety of tasks including density estimation, regression, and classification, on settings including asynchronous time sequences, images, tabular, and text data.
>
>
>
> > Does the adaptive strategy bring extra training cost? More detailed discussion is required.
>
> In Section 4.5, we provided detailed discussion about the wall-clock time required to find the adaptive flood levels. Once we have these levels, there is essentially no computational overhead; adding one absolute value to the loss function is negligible.
>
> If we use $M$ auxiliary networks, the time overhead is roughly $\mathcal{O}(M T)$, where $T$ is the training time of each auxiliary model. However, the efficient fine-tuning variant can reduce this to $\mathcal{O}(T + M t)$, where $t \ll T$ is the amount of time to fine-tune a model, without losing performance. Furthermore, since these adaptive flood levels are not closely tied to the “downstream” model, we can reuse them for a variety of downstream tasks, as was done e.g. in the C-score paper [1]. We will release these adaptive flood levels as a public difficulty measure, along with the code.
>
> Another important point is that AdaFlood models converge faster than unregularized or Flood-based models. For instance, on CIFAR10, AdaFlood’s validation loss converges after around 80 epochs, compared to 150 for unregularized models, 150 for Flood, and 100 for iFlood. Similarly, on CIFAR100, while AdaFlood and iFlood’s validation loss converges after around 60 epochs, unregularized and Flood take around 150 epochs to converge. Thus, once we have adaptive flood levels, a significant amount of training time can be saved. This effect will be amplified as it is amortized for multiple training runs, e.g. when others use our publicly-released flood values.
>
> [1] Ziheng Jiang, Chiyuan Zhang, Kunal Talwar, and Michael C Mozer. “Characterizing structural regularities of labeled data in overparameterized models”. ICML 2021.

---

### Author Response · Authors · 2023-11-20
**Common Response (2/2)**

> Additional experiments on large-scale dataset

We first want to mention that while we acknowledge the importance of dataset size, our primary focus has been to highlight the versatility of our proposed method. We have demonstrated its effectiveness across various tasks - density estimation, regression, and classification - spanning asynchronous time sequences, images, tabular, and text datasets. We hope that reviewers weigh more on the method's versatility rather than solely considering its scale, especially in the context of image classification.

As requested by multiple reviewers, we conducted additional experiments on larger scale datasets. Although Reviewer R2kv suggested ImageNet and vFax suggested ImageNet and LAION, which contain 1.2M and 400M images, respectively, due to limited time and resources we instead conducted image classification experiments on ImageNet100 (a subset of 100 classes from ImageNet, provided by [1]), and tabular regression on the NYC Taxi Tip dataset (one of the largest tabular datasets found in Grinsztajn et al. [2] with $>500,000$ rows). Given that ImageNet100 contains more than $100,000$ images, we believe it is a large enough proxy to establish the scaling behavior of AdaFlood.

The table below compares flooding methods on ImageNet100, both on the original dataset and on a version where 30% of data points have been deliberately mislabeled 30% (as in Section 4.3). We report test accuracies, along with expected calibration error (ECE) in parenthesis. Although Flood and iFlood do not improve the performance over the unregularized model, AdaFlood improves the performance by about 0.80% over the unregularized baseline: an even larger gap than than we observed for SVHN and CIFAR. We conjecture this is because ImageNet contains more noisy samples: it is well-known that many ImageNet images contain multiple objects, although the label is based on only one. This gap increases further to 1.1% in the presence of label noise, which again demonstrates the robustness of AdaFlood.

We are currently working on the full ImageNet dataset. With our limited resources, it will take another few days to finish. We will add the result in the main body of the revised paper. For now, we added the ImageNet100 result to Appendix C.


Methods | Mislabeled Samples 0% | Mislabeled Samples 30% |
| -------------- | ------------------ | -------------------- |
Unreg. | 81.00 (6.64) | 68.12 (17.23) |
Flood | 81.18 (6.44) | 68.19 (17.71) |
iFlood | 81.04 (6.86) | 68.24 (20.67) |
AdaFlood | 81.79 (4.81) | 69.22 (17.45) |

Table. Comparison of flooding methods on ImageNet100 with and without label noise.

The table below compares flooding methods on NYC Taxi Tip dataset with and without noise (the same type of noise as in Section 4.3). We report mean square error (MSE) and R2 score in parenthesis. Note that R2 score is usually between 0 and 1, but can be negative when predictions are bad. All flooding methods perform similar to the unregularized baseline without the presence of noise. The same is true for Flood and iFlood in the noisy settings, but AdaFlood significantly outperforms the other methods when noise is added. In particular, while R2 scores of other methods go below 0, it does not happen with AdaFlood, which demonstrates the robustness of AdaFlood even for the large-scale dataset like NYC Taxi Tip.

Methods | Noise Level 0.0 | Noise Level 1.5 | Noise Level 3.0 |
| -------------- | ------------------ | -------------------- | ------------------ |
Unreg. | 0.2373 (0.3335) | 0.3707 (-0.0409) | 0.3910 (-0.0978) |
Flood | 0.2373 (0.3335) | 0.3707 (-0.0409) | 0.3904 (-0.0980) |
iFlood | 0.2370 (0.3374) | 0.3652 (-0.0255) | 0.3902 (-0.0986) |
AdaFlood | 0.2369 (0.3348) | 0.3520 (0.0250) | 0.3465 (0.0119) |

Table. Comparison of flooding methods on NYC Taxi Tip tabular dataset with and without label noise.


We added these results, including detailed implementation information, in Appendix C.

[1] Yonglong Tian, Dilip Krishnan, and Phillip Isola. “Contrastive multiview coding”. ECCV 2020.

[2] Léo Grinsztajn, Edouard Oyallon, and Gaël Varoquaux. “Why do tree-based models still outperform deep learning on typical tabular data?”. NeurIPS 2022.

---

### Author Response · Authors · 2023-11-20
**Common Response (1/2)**

We appreciate all the reviewers’ valuable feedback. In this common response, we address the comments that were raised by multiple reviewers. Other responses are provided to individual reviewers. All the changes are reflected in the revised version, in blue font. If the reviewers have any concerns or questions, we are eager to discuss and answer them further.

Before responding to any comments, we would like to remind the reviewers of the overall contributions of our paper:

- Flood and iFlood provide surprisingly simple yet effective strategies to reduce overfitting by adding a constant flood level as regularization. However, they did not consider that training samples are not uniformly difficult. We explore this difference in the difficulty of training samples in Section 3.1.
- To address this issue, we propose Adaptive Flooding (AdaFlood) in Section 3.2, along with an efficient variant in Section 3.3. We also provide theoretical support for AdaFlood in Section 3.4.
- We demonstrate the effectiveness of AdaFlood for density estimation on asynchronous time series data in Section 4.1 and image classification in Section 4.2. Furthermore, we show the robustness of AdaFlood in noisy settings in Section 4.3, and that it gives better-calibrated models in Section 4.4.

---

### Author Response · Authors · 2023-11-22
**Discussion Ends in about a Day**

Dear Reviewers,

We again deeply appreciate your constructive feedback. We have carefully addressed all your comments and suggestions, making necessary revisions to our paper. With the discussion deadline approaching in about a day, we kindly remind the reviewers to consider our responses and re-evaluate our work. We are available for any additional questions or concerns, eager to engage in further discussion. Your feedback is invaluable to improving the quality of our work. We look forward to hearing from you soon. Thank you.

---

### Author Response · Authors · 2023-11-22
**Discussion Ends in about 12 hours**

Dear Reviewers,

As the deadline for the discussion is fast approaching, we want to ensure that all of your concerns have been addressed. We would greatly appreciate any feedback you may have regarding our responses. Thank you for your time and consideration.

---

### Meta-Review · Area_Chair_MjDm · 2023-12-06

**Metareview:**

All the reviewers provided the negative rating, where the main concerns are: 1) experiments on large-scale datasets and other learning schemes (e.g., self-supervised learning); 2) some theatrical validations and technical clarifications; 3) improvement over baselines. After rebuttal, some of the concerns are resolved by the authors. While the paper indeed has some merits, the AC took a close look at the paper and agrees with some remaining concerns, e.g., whether the proposed method can be effective in various learning schemes, how to measure the instance-wise difficulty. Hence, the rejection rating is recommended.

**Justification For Why Not Higher Score:**

While the paper tackles an interesting problem and provided more experimental results on more datasets in the rebuttal, the authors would still need more efforts to address all the reviewers' concerns.

**Justification For Why Not Lower Score:**

N/A

---

### Decision · Program_Chairs · 2024-01-16

Reject